# The Association between Frailty and Disability among the Elderly in Rural Areas of Korea

**DOI:** 10.3390/ijerph16142481

**Published:** 2019-07-11

**Authors:** Yeun-Soon Choi, Mi-Ji Kim, Gyeong-Ye Lee, Young-Mi Seo, Ae-Rim Seo, Bokyoung Kim, Jun-Il Yoo, Ki Soo Park

**Affiliations:** 1Department of Preventive Medicine, Institute of Health Sciences, College of Medicine Gyeongsang National University, Jinju-si, Jinju 52727, Korea; 2Center for Farmer’s Safety and Health, Gyeongsang National University Hospital, 79 Gangnam-ro, Jinju-si, Jinju 52727, Korea; 3Department of Orthopedic Surgery, Gyeongsang National University Hospital, 90 Chiram-dong, Jinju, Gyeongsangnam-do, Jinju 52727, Korea

**Keywords:** disability, elderly, frailty, rural

## Abstract

Background: The rapid increase in the elderly population in Korea is associated with an expanded burden of health problems. The purpose of this study was to investigate the association between frailty and physical disability among Koreans using the frailty index, which was developed to assess health conditions in elderly people. Methods: We included 503 elderly people from the Namgaram-II cohort. We used the Korean version of the Kaigo-Yobo checklist as our frailty assessment tool. For the disability assessment tool, we used the Korean version of the 12-item World Health Organization Disability Assessment Schedule (WHODAS-12). We fit multiple linear regression models for men and women for each section. Our models also included variables for musculoskeletal diseases that are known to be associated with frailty, such as sarcopenia, osteoporosis, and radiologic knee osteoarthritis. Results: After correcting for social demographic characteristics, blood profiles, high blood pressure, and diabetes, the Kaigo-Yobo results showed a significant difference in frailty between men (1.53 ± 0.74) and women (2.60 ± 0.77), and WHODAS-12 also showed a significant difference between men (6.59 ± 5.08) and women (15.99 ± 5.70). After correcting for social demographic characteristics, blood profiles, high blood pressure, and diabetes, Kaigo-Yobo and WHODAS-12 were significantly associated with each other among both men (β = 2.667) and women (β = 3.200) (*p* < 0.001). Conclusions: The study results show an association between frailty and disability among elderly people in rural areas. Therefore, prevention should occur at the pre-frailty stage of a person’s life to prevent further disability. Also, disability welfare programs should be provided to elderly people who present with frailty.

## 1. Introduction

In 2000, Korea officially became an aging population as the ratio of people who were 65 years or older exceeded 7% of the total population; in 2017, this proportion reached 14.2%. In rural areas, the population began aging more rapidly, resulting in a “super-aged” population [1]. 

As Korea moves toward becoming a super-aged population, frailty has become a critical issue. Frailty increases weaknesses and decreases the ability to sustain consistency, which results in clinical syndromes like weight loss, fatigue/exhaustion, weakness, low physical activity, mobility impairment, and declining physical performance. These syndromes lead to frequent falls and hospital visits, which ultimately lead to disability and death [2,3]. Fried et al. [2] indicated that the criteria for frailty according to clinical phenotype include three out of five of the following syndromes: weight loss, fatigue/exhaustion, weakness, slowness, and low physical activity. Clinical phenotypes are often accompanied by psychological syndromes such as cognitive impairment or depression [2,3,4].

The physical and cognitive changes experienced by the elderly can cause them to reduce their interactions with other people, resulting in isolation and a deterioration in social skills [5]. Social frailty also interacts with physical and cognitive frailty and affects health outcomes. While many previous studies classified frailty into separate categories including physical, cognitive, and psychological [1,4,6,7,8], studies that explore comprehensive health issues, including social variables, are rare.

Historically, disability definitions were limited and only concerned physical ability, but a new comprehensive definition by the International Classification of Function, Disability, and Health (ICF) [9] includes social and psychological domains, such as functional impairment, activity limitation, and social participation restriction. In accordance with this definition, the World Health Organization Disability Assessment Schedule (WHODAS) questionnaire was created [10]. WHODAS was subsequently created to cover physical, psychological, and social disability, and it is widely used globally [11,12,13,14,15].

Frailty studies mostly focus on the association between frailty and disability for daily living, which is measured by the Active Daily Living (ADL) and Instrumental Activities of Daily Living (IADL) indices, among others [1,6,16]. There are currently no comprehensive disability-related studies that cover the physical, psychological, and social aspects of frailty.

Therefore, the purpose of this study was to investigate the association between frailty and physical disability using the frailty index, which was developed to assess the health conditions of elderly people. Another objective of this study was to test whether disability is also associated with osteoarthritis, sarcopenia, and osteoporosis, which all affect disability, and if any of these disorders are still associated with disability even after accounting for potential confounding variables. 

## 2. Materials and Methods

### 2.1. Subjects

The Namgaram-II cohort has been recruiting participants since 2016 to follow rural farmers who receive medical check-ups for musculoskeletal disorders; Namgaram-II targets elderly subjects who are 65 years or older. This cohort has been described in previous research. Cohort subjects underwent cognitive evaluation (using the Mini-Mental State Examination—Dementia Screening, MMSE-DS), and, using age and education as cut-off markers, we identified and included subjects with no cognitive impairment [17]. All the surveys were conducted on a one-on-one basis after trained researchers screened the subjects, explained the survey contents, and obtained written consent. This study was approved by the Institutional Review Board of Gyeongsang National University (approval number: GIRB-A16-0012).

### 2.2. Research Tools and Methods

Surveys elicited socio-demographic data, including sex, age, marital status, and subjective economic status. As is known, questions about income in elderly are hard to answer, missing data are more likely, the chance of selection bias increases, and, as a result, the interpretation of results could be underestimated. As an alternative, a subjective question about economic status is less intrusive, participants are less reluctant to answer, and missing data are reduced [18]. Subjective economic status was divided into five categories for living expenses: very unsatisfied, unsatisfied, average, satisfied, and very satisfied. The five subjective economic categories were then aggregated into three categories: unsatisfied, average, and satisfied. Subjects were considered to be patients only when they reported taking medical treatment for high blood pressure or diabetes.

Blood tests included determination of hemoglobin, albumin, cholesterol, and 25(OH) VitD, which are associated with nutrition status. Radiologic knee osteoarthritis (RKOA), osteoporosis, and sarcopenia, all of which are musculoskeletal disorders, were also evaluated.

FrailtyThe Kaigo-Yobo checklist [19] was used to measure frailty because it has been evaluated to be a reliable and valid survey tool that covers social activity and daily living for elderly Korean subjects [20]. Kaigo-Yobo is a frailty measurement tool that combines domain and expression variables. Out of a total of 15 items, 4 evaluate nutrition status, 3 evaluate falling incidents, 2 evaluate tendency to take walks, 2 evaluate social relations and assistance from family or friends, and 1 item evaluates subjects’ general health condition, communication, ability to move around, and enjoyment of hobbies. All answers are binary (yes or no), and 1 point is added for each item, adding up to a potential highest score of 15; the higher the score, the greater the subject’s frailty.DisabilityThe World Health Organization (WHO) developed the WHODAS, which evaluates concepts from the International Classification of Functioning, Disability, and Health (ICF) to measure individual functioning and disability across a range of aspects [10]. WHODAS measures difficulty in six domains of daily life (cognition, mobility, self-care, getting along, life activities, and participation in social activities) and difficulty following disease onset, damage, and health conditions over short or long time periods. For this study, we adopted the WHODAS-12, Korean version [11], which has been shown to be reliable and valid. WHODAS-12 requires responses ranging from “no difficulty” (1) to “extreme difficulty” (5) and then asks subjects how difficult it has been to carry out the specific task over the last month. The total score can range from 0 to 100 points; higher scores represent more severe disability.Measurement of Musculoskeletal Disease(1) SarcopeniaDual-energy X-ray absorptiometry (DEXA; Discovery W, Hologic, Waltham, MA, USA) was used to measure the limb skeletal muscle index (SMI), which is obtained by dividing the appendicular skeletal mass (ASM) by the subject’s squared height (SMI = ASM/Ht²). For men, sarcopenia is defined as SMI < 7.0 kg/m^2^; for women, it is SMI < 5.4 kg/m^2^ [21].(2) Radiologic Knee Osteoarthritis (RKOA)Radiographic images of both knees were taken, and RKOA was defined as Level 2 or higher for at least one joint on the Kellgren–Lawrence (K/L) classification system. Radiographic images were read by two radiologists, each with more than 20 years’ experience in musculoskeletal evaluation at a teaching hospital. When their opinions conflicted, they discussed their opinions until an agreement was reached.(3) OsteoporosisUsing DEXA (Discovery W, Hologic), the bone density in the lumbar area was measured. A T score of −2.5 or less was defined as osteoporosis.

## 3. Data Analysis

All data were analyzed using SPSS 23.0 software (IBM, Armonk, NY, USA). Subjects’ general characteristics are reported in descriptive analyses, among which continuous variables are reported as averages and standard deviations and categorical variables as numbers and percentages. We conducted a test to check if there was a difference in the average Kaigo-Yobo and WHODAS II scores between men and women, for which we considered numerous demographic variables (age, gender, marital status, economic status), high blood pressure and diabetes, and pre- and post-correction for sarcopenia, osteoporosis, RKOA, and blood tests (hemoglobin, albumin, cholesterol, 25(OH)VitD). Correlation analyses showed an association between Kaigo-Yobo and WHODAS-12. To verify the association between Kaigo-Yobo and WHODAS-12, we fit two multiple linear regression models. Model 1 was adjusted for social demographic variables (age, sex, marital status, financial status), high blood pressure and diabetes, and blood tests (hemoglobin, albumin, cholesterol and 25(OH)VitD). Model 2 included all variables in Model 1 as well as musculoskeletal disorder variables (sarcopenia, osteoporosis, RKOA).

## 4. Results

### 4.1. Subject Characteristics

In total, we recruited 503 subjects aged 65 years or older, including 339 women (67.4%), and the sample had a mean age of 72.7 ± 5.0 years (range: 65–86 years). Of the subjects, 295 (58.6%) were currently married, and 169 (33.6%) replied that they were satisfied with their economic status. Regarding disease, 282 subjects (43.9%) were diagnosed with high blood pressure, 132 (26.2%) had diabetes, 96 (19.4%) had osteoporosis, 137 (28.5%) had RKOA, and 89 (17.7%) had sarcopenia (Table 1).

### 4.2. Score Differences between Kaigo-Yobo and WHODAS-12 by Sex

Before correcting for confounding variables, average Kaigo-Yobo scores were significantly different between men (1.51 ± 1.63) and women (2.60 ± 2.02; *p* < 0.001). After correcting for socio-demographic variables (age, marital status, subjective economic status); high blood pressure; diabetes; blood tests (hemoglobin, albumin, cholesterol, and 25(OH)VitD); and sarcopenia, osteoporosis, and RKOA, the men’s average was 1.53 ± 0.74 and the women’s was 2.60 ± 0.77, which was still significantly different (*p* < 0.001; Figure 1).

For WHODAS-12, before correction, the men’s average was 6.52 ± 9.48 and the women’s was 16.16 ± 13.58, which was a significant difference (*p* < 0.001). After correcting for the above confounding variables, the men’s average was 6.59 ± 5.08, and the women’s was 15.99 ± 5.70, which was again a significant difference (*p* < 0.001; Figure 2).

### 4.3. Correlation between WHODAS-12 and Kaigo-Yobo

We conducted a correlation analysis between Kaigo-Yobo and WHODAS-12, according to sex. The men’s correlation coefficient value was 0.494 (*p* < 0.001) and the women’s was 0. 536 (*p* < 0.001); both scores were significantly correlated among men and women (Figure 3).

To verify individual correlations between disability and frailty, multiple linear regression analysis was conducted and stratified by sex. For men, after adjusting for age, spouse, subjective economy status, diabetes, hypertension, hemoglobin, albumin, total cholesterol, and 25(OH) vitamin D (Model 1), WHODAS-12 was significantly associated with Kaigo-Yobo (β = 3.109, *p* < 0.001). Even after adjusting for the variables of Model 1 and sarcopenia, osteoporosis, and radiologic knee osteoarthritis (Model 2), WHODAS-12 was significantly associated with Kaigo-Yobo (β = 2.667, *p* < 0.001). The coefficients of determination (R^2^) were 0.421 (*p* < 0.001) in Model 1 and 0.479 (*p* < 0.001) in Model 2; the change in R^2^ between Model 1 and Model 2 (Δ = 0.058) was statistically significant (*p* = 0.001). 

For women, both Model 1 (β = 3.206, *p* < 0.001) and Model 2 (β = 3.200, *p* < 0.001) showed a significant correlation with Kaigo-Yobo and WHODAS-12. The coefficients of determination (R^2^) were 0.369 (*p* < 0.001) in Model 1 and 0.376 (*p* < 0.001) in Model 2; the change in R^2^ between Model 1 and Model 2 (Δ = 0.007) was not statistically significant (*p* = 0.202) (Table 2). 

## 5. Discussion

As the average life expectancy of the elderly continues to rise, so does concern about frailty. Frailty is related to hospitalization, falls, disability, and, eventually, death. Thus, this study analyzed the association between frailty and disability in elderly rural-living farmers who were 65 years or older, after correcting for differences in social demographic variables, diabetes, high blood pressure, blood test, osteoporosis, knee osteoarthritis, and sarcopenia. Our results suggest that there is a significant association between frailty and disability in this population. 

Many studies that have reported on physical disabilities measured frailty with the Activities of Daily Living (ADL) and Instrumental Activities of Daily Living (IADL) indices [2,6,8,16,19,21,22]. Meanwhile, other studies have identified an association between frailty and cognitive impairment [1,23,24]. Additionally, some studies have evaluated the association between frailty and not only physical function and cognitive skill, but also participation in social activities. Gutierrez [25] outlined how social skills, life habits, and physical vulnerability can affect frailty. Bunt et al. [5] explained how social skills and physical and psychological function can all decline in the manner of a vicious cycle. They further explained that malfunction in social skills, such as reduced communication with others, can lead to reduced appetite, worse nutrition, higher risk of falls and fear of going out for a walk, hindered communication opportunities, and eventual worsened physical function. Accordingly, frailty is closely associated with disability.

However, previous studies on frailty and disability mostly focused on individuals, and they separated aspects of physical function, cognitive skill, and social activity. The WHO International Classification of Functioning, Disability and Health (ICF) defined disability as “difficulty in carrying out functions at an individual or social level,” [9] and they subsequently developed WHODAS as a comprehensive measure of disability [10]. WHODAS is already established as a widely used, internationally standardized measuring tool that can be compared across countries [12]. It also offers the advantage of simultaneously measuring disability multidimensionally. This study used the WHODAS tool to comprehensively cover the physical, psychological, and social aspects of disability. Even though we made corrections to the sarcopenia and RKOA measurements, which are known to be associated with frailty and disability and to severely affect frailty in particular, associations between frailty and disability were present among both sexes. In other words, even though objective measurements and corrections were made to a musculoskeletal disorder that is known to be associated with frailty, it was still associated with frailty and disability. This outcome is meaningful because it redresses the insufficient objectivity of the K-Y index and WHODAS II, which consist of questionnaires only.

Regarding sarcopenia, the effects of muscle weakness on disability have been found to be different between men and women. Fragala et al. [26] performed a prospective study using 470 older adults to evaluate the difference in anthropometric predictors of physical performance in older women and men. They reported that muscle quality in men was more important for functional performance than in women. Their findings about how sarcopenia affects physical disability in men are consistent with ours in this study. The difference between men and women is an important finding that emphasizes early screening of muscle loss in rural populations.

This study has several limitations. First, it was difficult to detect a dynamic cause-and-effect relationship between disability and frailty because of the cross-sectional design. Second, there is a high risk of bias in the sample, and the sample population was too narrow to generalize to broader populations because the subjects all lived in a limited local area. We will attempt to address these insufficiencies in future research. 

## 6. Conclusions

A local health program for people living in remote and rural areas is necessary to decrease frailty and reduce the risk of irreversible disability and early death among the elderly. Social welfare and health programs for the elderly should be introduced, given the insight that frailty leads to disability. 

## Figures and Tables

**Figure 1 ijerph-16-02481-f001:**
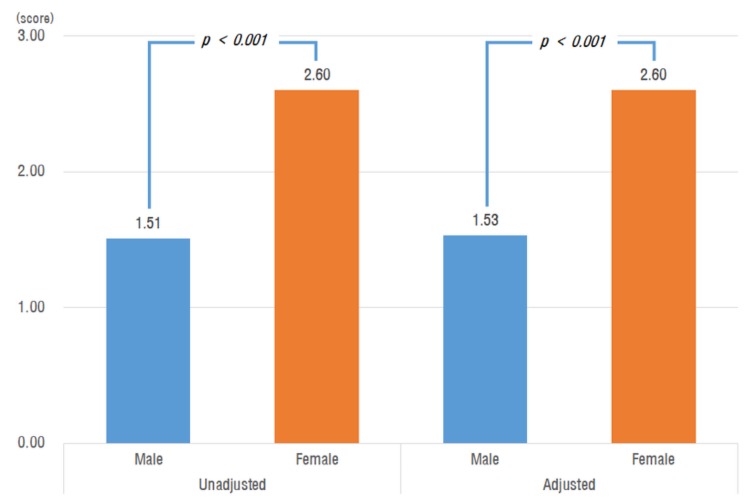
Kaigo-Yobo score by sex; *p*-value from *t*-test; Adjusted for age, sex, spouse, subjective economy, hypertension, diabetes mellitus, sarcopenia, osteoporosis, RKOA, Hb, total cholesterol, albumin, and 25(OH) vitamin D.

**Figure 2 ijerph-16-02481-f002:**
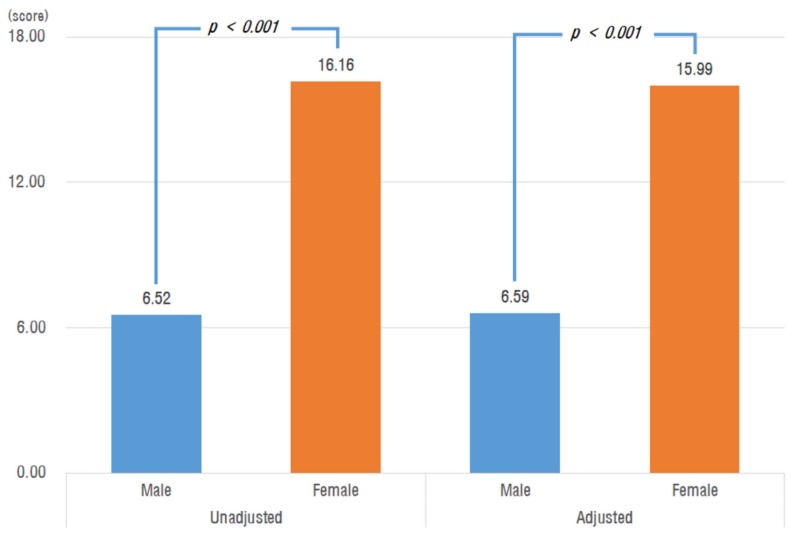
World Health Organization Disability Assessment Schedule-12 (WHODAS-12) score by sex; *p*-value from *t*-test; Adjusted for age, sex, spouse, subjective economy, hypertension, diabetes mellitus, sarcopenia, osteoporosis, RKOA, Hb, total cholesterol, albumin, and 25(OH) vitamin D.

**Figure 3 ijerph-16-02481-f003:**
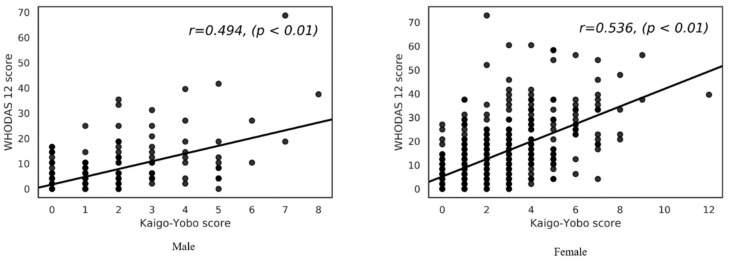
Association between Kaigo-Yobo and World Health Organization Disability Assessment Schedule-12 (WHODAS-12).

**Table 1 ijerph-16-02481-t001:** Subject characteristics by sex (*n* = 503).

Characteristics	Categories	Total		Male		Female	
		*n*	%	*n*	%	*n*	%
Sex	Male	164	32.6				
	Female	339	67.4				
Mean age (years) ± SD		72.7 ± 5.0		71.6 ± 4.9		73.2 ± 4.9	
Spouse	Yes	295	58.6	148	90.2	147	43.4
	No	208	41.4	16	9.8	192	56.6
Subjective economic status	Low	169	33.6	43	26.2	126	37.2
	Middle	239	47.5	80	48.8	159	46.9
	High	95	18.9	41	25.0	54	15.9
Hypertension	Yes	282	43.9	80	48.8	202	59.6
	No	221	56.1	84	51.2	137	40.4
Diabetes	Yes	132	26.2	46	28.0	86	25.4
	No	371	73.8	118	72.0	253	74.6
Osteoporosis	Yes	405	80.6	154	93.9	251	74.1
	No	98	19.4	10	6.1	88	25.9
RKOA	Yes	137	28.5	146	90.7	197	61.8
	No	343	71.5	15	9.3	122	38.2
Sarcopenia	Yes	89	17.7	19	11.6	70	20.6
	No	414	82.3	145	88.4	269	79.4
Total		503	100	164	100	339	100

Data are presented as means ± SDs and proportions (%). RKOA: Radiologic Knee Osteoarthritis.

**Table 2 ijerph-16-02481-t002:** Regression analysis for World Health Organization Disability Assessment Schedule 12 (WHODAS-12).

Variables	Male								Female							
	Model 1				Model 2				Model 1				Model 2			
	Nonstandard Coefficient		Standard Coefficient	*p*-value	Nonstandard Coefficient		Standard Coefficient	*p*-value	Nonstandard Coefficient		Standard Coefficient	*p*-value	Nonstandard Coefficient		Standard Coefficient	*p*-value
	B	SE	β		B	SE	β		B	SE	β		B	SE	β	
K-Y score	3.109	0.393	0.536	< 0.001	2.667	0.401	0.461	< 0.001	3.206	0.315	0.476	< 0.001	3.200	0.344	0.461	< 0.001
Sarcopenia					7.658	2.072	0.254	< 0.001					0.243	1.663	0.007	0.884
(yes/no)																
Osteoporosis					2.363	2.512	0.060	0.348					−2.483	1.544	−0.080	0.109
(yes/no)																
RKOA					3.205	1.997	0.098	0.111					1.793	1.344	0.064	0.183
(yes/no)																
	Adjusted R^2^ = 0.421 (*p* < 0.001)				Adjusted R^2^ = 0.479 (*p* < 0.001)				R^2^ = 0.369 (*p* < 0.001)				R^2^ = 0.376 (*p* < 0.001)			
	ΔR^2^ = 0.058 (*p* = 0.001)								ΔR^2^ = 0.007 (*p* = 0.202)							

Adjusted for age, spouse, subjective economy, diabetes, hypertension, hemoglobin, albumin, total cholesterol, and 25(OH) Vitamin D; K-Y score: Kaigo-Yobo score; RKOA: Radiologic Knee Osteoarthritis.

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
