# Peer review of "The Association between Frailty and Disability among the Elderly in Rural Areas of Korea"

_ijerph, 2019, doi:10.3390/ijerph16142481_

Round 1
Reviewer 1 Report
Interesting paper. I wonder if the authors could add either the Multiple R or R-squared values for the regression equations discussed. It would help the reader understand the strength of the multivariate relationships. Also, on line 172 - could the authors briefly describe Models 1 and 2. I recommend publication
Author Response
About the Multiple R or R-squared values for the regression equations discussed(line 175-186). Also, on line 172 - could the authors briefly describe Models 1 and 2.
--- The coefficient of determination(R2) was presented at the table 2 and discussed in the text..
----I described Models 1 and 2 on the test (yellow highlighted)
Line-175-186
To verify individual correlations between disability and frailty, multiple linear regression analysis was conducted and stratified by sex. For men, after adjusting for age, spouse, subjective economy status, diabetes, hypertension, hemoglobin, albumin, total cholesterol, 25(OH) vitamin D (Model 1), WHODAS-12 was significantly associated with Kaigo-Yobo (B=3.109, p<0.001). Even after adjusting for variables of model 1 and sarcopenia, osteoporosis and radiologic knee osteoarthritis (Model 2), WHODAS-12 was significantly associated with Kaigo-Yobo (B=2.667, p<0.001). Coefficient of determination (R2) was 0.421 (p<0.001) in model 1, 0.479 (p<0.001) in model 2, respectively, R2 change between model 1 and model 2 (Δ=0.058) was significant statistically(p=0.001).
And for women, both model 1 (B=3.206, p < 0.001) and model 2 (B=3.200, p < 0.001) showed a significant correlation with Kaigo-Yobo and WHODAS-12. Coefficient of determination (R2) was 0.369 (p<0.001) in model 1, 0.376 (p<0.001) in model 2, respectively, R2 change between model 1 and model 2 (Δ=0.007) was not significant statistically (p=0.202) (Table 2).
Reviewer 2 Report
Review of the manuscript: The Association between Frailty and Disability Among the Elderly in Rural Areas of Korea
Manuscript Number: IJERPH-545406
The study was to investigate the association between frailty and physical disability among Koreans using the frailty index, which was developed to assess health conditions in elderly people.
1. Formal comments
Adequacy for the audience of the magazine: the article (for its theme as for its expository form) is suitable for the audience of the magazine.
Originality of work in your field: The text is original and of methodological interest. It deals with a central issue in exclusion studies, namely frailty in elderly people, associated with social and health processes.
Title and summary: The title is attractive and informs of the purpose of the text. The summary is complete and adjusted to academic standards.
Formal aspects: the text is easy to read. The contents are well ordered: Summary, Introduction, Materials and Methods, Data Analysis, Results and Discussion, in addition to the bibliography.
The tables are perfectly understandable and complement the information in the Analysis of Data and Results section. The extension of the text is adequate.
2. Comments on the academic / scientific content of the text
Introduction and objective: The objective is clearly defined from the beginning and a brief introduction is made that contextualizes the study well.
Methods: the procedure has enough sample and the study design is correct. The sample is well selected, with clear exclusion / inclusion criteria. It seems a good option to exclude subjects with cognitive impairment from the study. In addition, part of a previously studied cohort, a fact that makes it possible to follow the subjects.
The study uses a social categorization criterion based on a deprivation indicator (Economic status was divided into five categories for living expenses: very unsatisfied, unsatisfied, average, satisfied, and very satisfied). Why was not a more objective indicator used: family income, individual income, probability of poverty risk ...? This issue should be clarified by the authors.
The choice of The Kaigo-Yobo checklist is adequate and reinforces the multidimensionality of the concept of frailty in the elderly.
The use of the WHODAS instrument is also adequate and is validated internationally.
Results: The relevant information is presented to answer the research question.
Discussion: It does not repeat the findings of the study. Raises relevant issues to follow the problem of frailty of the elderly
Author Response
The study uses a social categorization criterion based on a deprivation indicator (Economic status was divided into five categories for living expenses: very unsatisfied, unsatisfied, average, satisfied, and very satisfied). Why was not a more objective indicator used: family income, individual income, probability of poverty risk ...? This issue should be clarified by the authors.
---I changed variable of economy into subjective economy status in the context and table, explained about it in the context(line 77-85), and added reference 18.
and subjective economic status. As is known, questions about income in elderly are hard to answer, missing data is more likely, the chance of selection bias increase and as a result, the interpretation of results could be underestimated. As an alternative, the subjective question about economic status is less intrusive and reluctant to answer and missing dates are reduced18. Subjective economic status was divided into five categories for living expenses: very unsatisfied, unsatisfied, average, satisfied, and very satisfied. The five subjective economic categories were then aggregated into three categories: unsatisfied, average, and satisfied. Subjects were considered to be patients only when they reported taking medical treatment for high blood pressure or diabetes